# Preventive Behaviors and Influencing Factors among Thai Residents in Endemic Areas during the Highest Epidemic Peak of the COVID-19 Outbreak

**DOI:** 10.3390/ijerph20032525

**Published:** 2023-01-31

**Authors:** Weerawat Ounsaneha, Orapin Laosee, Thunwadee Tachapattaworakul Suksaroj, Cheerawit Rattanapan

**Affiliations:** 1Faculty of Science and Technology, Valaya Alongkorn Rajabhat University under the Royal Patronage, Klong Nuang, Klong Luang, Pathum Thani 13180, Thailand; 2ASEAN Institute for Health Development, Mahidol University, Salaya, Phutthamonthon, Nakhon Pathom 73710, Thailand

**Keywords:** preventive behavior, Thai residents, epidemic peak, COVID-19

## Abstract

This research aims to investigate COVID-19 preventive behavior and influencing factors among Thai residents during the highest epidemic peak of COVID-19. Nine hundred and forty-six residents in five districts with high COVID-19 infection cases in Thailand were systematically included in this cross-sectional survey. The results showed that 87.2% and 65.2% of the residents had a high level of general knowledge and preventive measures, respectively. As to COVID-19 attitudes, poor levels of attitude among Thai residents were found in risk perception (53.6%) and mistrust issues (70.4%). Moreover, this study presents good preventive behavior (77.0%) among Thai residents. Multiple logistic regression showed that the influence factors of COVID-19 preventive behavior were the young age group (AOR 2.97, 95% CI 1.68–5.25), high income (AOR 1.38, 95% CI 1.03–1.86), and high level of general COVID-19 knowledge (AOR 2.21, 95% CI 1.64–2.96). The conclusion was that providing information on COVID-19 via social media was the key mechanism of policy action for increasing the level of COVID-19 preventive behavior during the highest epidemic peak in Thailand. In addition, the pandemic preparedness and response policy, with resident participation and involvement, could be recommended for the resilience of pandemic preparedness.

## 1. Introduction

In late December 2019, a novel coronavirus (COVID-19) outbreak was first reported in Wuhan city, China, with death cases of about 1800 and infection cases of more than 70,000 in the first 7 weeks of the epidemic [1]. In March 2020, a new global pandemic situation was formally announced by the World Health Organization (WHO), recommending that the world should actively respond to the outbreak [2]. The COVID-19 epidemic spread across the world at a high capacity of transmission, which differed from other coronaviruses [3]. Lakshmi and Suresh [3] predicted that between 9 and 22 million people died during the COVID-19 period. The high-risk population, including people with obese status, patients with cancer, smokers, and those with chronic diseases, heart disease, and kidney disease, paid serious attention to reducing and preventing COVID-19 transmission [4,5]. To contain the outbreak, prevention and control guidelines, including face mask use, hand washing and cleaning, and social distancing, were recommended to protect the populations and decrease virus exposure [6]. However, adaptation to the “new normal” behaviors was limited [7], and the stressful situation of COVID-19 spread caused people to ignore these preventive behaviors [8].

The first country outside of China to detect COVID-19 was Thailand [9]. COVID-19 in Thailand was declared by the Communicable Diseases Act B.E. 2558 as a dangerous communicable disease [10]. On 31 January 2022, 2,425,412 cases of COVID-19 and 22,173 deaths, with a 0.91% mortality rate, were reported by the Department of Disease Control, Ministry of Public Health, Thailand [11]. In addition, a high epidemic rate of COVID-19 confirmed cases was found in the Bangkok Metropolitan Area, which has a high population density [11]. In the beginning, the Ministry of Public Health suggested that the key obstruction and preventive measures consist of educating people regarding screening and self-protection with COVID-19-positive cases and searching for and promoting proactive public behavior [12]. Consequently, various COVID-19 treatment strategies and vaccine development began at the initial stage of research and study without an understanding of the certain consequences of the disease [13]. As a result, the following approaches to and policies of virus outbreak prevention were initiated and launched by Thailand and countries around the world: state quarantines, community involvement, border closures, travel bans, etc. [13,14].

Zhong et al. [15] implied that the factors associated with COVID-19 preventive behavior were knowledge, attitudes, and practices. Related reports [16,17] mentioned that poor knowledge of infectious diseases was associated with peoples’ negative emotions toward preventing disease transmission [16,17]. In addition, Park and Oh [18] identified that the high risk and severity perception of disease among people were involved in the important step of preventive behavior. The COVID-19 preventive behavior was significantly impacted by the threat of perception because it stimulated individuals to protect themselves from ongoing threats.

More than 25,000 COVID-19 patients were reportedly admitted to hospitals, constituting the highest epidemic peaks in Thailand from 27 March 2022 to 2 April 2022 [11]. Moreover, a higher number of deaths (more than 80 people/day) occurred during this epidemic peak. The results from Sasaki and Ichinose [19] reported that the effective reproduction of a number of viruses (the average virus number of secondary cases by an individual infection) was likely to be higher in a large urban setting because of the greater number of reproduction opportunities afforded by population densities and built-up environments [20,21]. Being the center of numerous social activities may be another reason for the high risk of viral transmission in urban areas. From the knowledge and experience obtained during the SARS outbreak in 2003, the correlation between infectious disease knowledge and the panic level in the population complicated efforts to prevent the spread of the disease [22].

However, there has been little work on the preventive behaviors involved in the highest epidemic peak of the COVID-19 outbreak. Therefore, an understanding of the influencing factors on COVID-19 preventive behavior could offer effective management of COVID-19 prevention and control in an extreme crisis situation. Relevant factors were determined concerning how to best act on and implement those required behaviors in the face of the panic outbreak [23,24]. This research aims to identify the association of personal factors, COVID-19 knowledge, attitudes, and preventive behaviors among residents during the highest endemic peak in Thailand. The survey of variables influencing COVID-19 preventive behavior provides key information backup to propose the most appropriate and effective mechanisms to decrease COVID-19 cases during extreme crisis situations.

## 2. Materials and Methods

### 2.1. Study Design and Setting

This is a cross-sectional study survey that was conducted from February 2022 to May 2022 to identify the knowledge level of COVID-19 preventive measures and their influencing factors among residents during the highest endemic peak in Thailand. The face-to-face interviews were used as the data collection technique that contained the questionnaires used in the present study.

The study setting areas were in five endemic provinces with the highest case rankings of COVID-19 infection in Thailand, from the data of the Ministry of Public Health [11], which included Bangkok, Samut Prakan, Chon Buri, Samut Sakhon, and Nonthaburi (Figure 1). The criteria for selecting study sites include: (1) the selection of five provinces in the urban area, (2) for each province, one district in each province was selected by a random sampling technique, and (3) the number of participants in the selected area should not be equal among the five provinces but by proportion to size (Figure 2). Five districts in each province were selected as areas for data collection: Thawi Watthana, Bang Bo, Bang Lamung, Krathum Baen, and Bang Yai. Finally, residents in five districts were sampled using a convenience technique to recruit the participants in this study.

### 2.2. Participants

The equation calculation from Cochran and Biswas [25] was used to estimate sample size with a 95% confidence level, an acceptable error of 30%, and a proportion of preventive behavior of 0.5. According to this calculation, 980 Thai residents were recruited as subjects, increasing by 25% the sample size. After missing data exclusion, 946 Thai residents from five provinces were the final sample size in this study. The inclusion eligibility criteria of residents were: aged over 18 years, Thai citizens, and living in an endemic setting for more than 6 months. Equal sample proportion, with gender and age group, was conducted for the residents’ recruitment process and the reduction of selection bias. Written informed consent was obtained by the well-trained research assistants from the participants who voluntarily agreed to enroll in the survey. The rights and privacy of data were strictly maintained. This research was approved by the Committee for Research Ethics (Social Science), Faculty of Social Science and Humanities, Mahidol University (certificate approval number 2022/033.2802 and MU-SSIRB number 2022/35(B2)).

### 2.3. Measurement

The measurement used in this study was a questionnaire divided into four parts. Part 1 included questions on general information, including age, gender, education level, marital status, occupation, monthly income, expenses per month, COVID-19 vaccine history, COVID-19 infection history, COVID-19 insurance, and COVID-19 information. Part 2 included 11 questions evaluating the knowledge of COVID-19; 1 point was given for correct answers and 0 points for incorrect answers. This part is divided into two items, following the knowledge of COVID-19 on general information and preventive measures, adapted from the results of Kamacooko et al. [26]. The levels of knowledge were categorized by total scores of low (0–79%) and high (80–100%) levels. Part 3 included 12 questions assessing the attitudes toward COVID-19, modified from the report of Masoud et al. [27]. A rating scale of five levels for each question was used in the questionnaire design in this part with two items: attitude toward risk perception and mistrust. Finally, Part 4 included 10 questions evaluating the preventive behavior of COVID-19, adapted from the report of Park Da-In et al. [28]. The level of preventive behavior of COVID-19 was evaluated by a five-level rating scale. The levels of attitude and prevention behavior toward COVID-19 were categorized by statistical mean scores.

### 2.4. Data Collection

After approval and ethical permission from the Committee for Research Ethics (Social Science), the pre-test questionnaire was used to perform the validity and reliability test of the data. Face-to-face interviews were conducted in the five endemic areas. Five research assistants were trained by the research team to ensure the understanding of the research measurement. The research team coordinated the health officers in each area of the health office, asking permission to conduct research in those areas. Because of the endemic peak of the COVID-19 situation in Thailand, the research team followed government regulations during the interview process of the research assistants in each study setting.

### 2.5. Statistical Procedures and Analysis

All the statistical analyses were performed using SPSS software version 21. Percentages, means or medians, standard deviations, or quartile deviations were applied for the data analysis in this research. The association between COVID-19 preventive behaviors and sociodemographic characteristics, knowledge, and attitudes was investigated by the chi-square test. Associations between the preventive behaviors and the independent variables, while simultaneously controlling for other confounding factors, were identified by multiple logistic regression analysis, with statistical significance at 0.05.

## 3. Results

### 3.1. Sociodemographic Characteristics of Residents

Of the 946 participants in the five endemic settings, 60.3% of female residents responded. The majority of the residents belonged to the age group of 25–37 years (30.2%) (Table 1). For the education level of residents, secondary school was the majority (33.2%), followed by a bachelor’s degree and above (26.7%), and then primary school (21.8%). Married residents were in the majority (52.2%), and 44.8% were single/divorced. In terms of occupation, 39.2% of residents were general employees, followed by the private sector and self-employed (23.3% and 17.7%, respectively). Of the residents who responded, 49.2% had a monthly income of 5001–10,000 Baht (BHT) (USD 150–300) per month. One-fourth of the residents (42.2%) had been vaccinated against COVID-19. Approximately five-tenths of the residents (53.2%) had reported a COVID-19 infection. Furthermore, family members were the majority sources (39.4%) of COVID-19 infection. A total of 81.2% of residents had no health insurance. 

### 3.2. Knowledge and Attitude Levels on COVID-19 Prevention

The levels of knowledge and attitude of the residents on COVID-19 were divided into two levels: low and high for knowledge of COVID-19 and poor and good for attitudes toward COVID-19. Table 2 shows the levels of knowledge and attitude toward COVID-19 infection. The COVID-19 knowledge levels among Thai residents were high in terms of general knowledge (87.2%) and preventive measures (65.2%). Among the 946 residents, 98.1% knew that isolating people infected with COVID-19 is an effective way to reduce the spread of the virus, and 97.5% knew that there is currently a symptomatic treatment and cure for COVID-19 (Table 3). For COVID-19 attitudes, poor levels of attitude among Thai residents were found in risk perception and mistrust issues. The majority of the residents (64.6%) were not sure if they believed that they would have low symptoms if they were infected with COVID-19. Around 62.8% of respondents strongly disagreed that people who have been infected with COVID-19 should not be condemned by society (Table 4).

### 3.3. Preventive Behavior Level toward COVID-19

The overall items of residents’ preventive behavior toward COVID-19 are shown in Table 5. The highest preventive behavior level (77.8%) for COVID was for wearing facial masks in public areas. More than 63% of residents reported washing their hands frequently and using soap or hand sanitizer in public areas prior to the interview. In the same period, reducing travel to public areas and working from home or having online meetings were the limited behaviors for COVID-19 prevention. However, 49.8% of residents followed all items of preventive behavior in this study. Overall, most of the residents (77%) presented good preventive behavior toward COVID-19.

### 3.4. Association between the Independent and Dependent Variables

Table 6 shows the statistical results from the chi-square test, logistic regression, and multiple logistic regressions, showing the association between the independent and dependent variables. The chi-square test was used to identify the variable associated with preventive behavior toward COVID-19. In bivariate analysis, younger age, high education level, high income, and a good level of COVID-19 knowledge tended to have good preventive behavior (*p*-value < 0.05). Statistical analysis indicated that residents of age less than 54 years, with a bachelor’s degree and above (COR 1.58, 95% CI 1.06–2.34), income > 10,000 THB/month (COR 1.50, 95% CI 1.13–1.98) and a high level of COVID-19 knowledge in general (COR 2.37, 95% CI 1.71–3.02) were significantly associated with preventive behavior. Multiple logistic regression analysis identified significant factors associated with preventive behavior toward COVID-19. In the younger age group (less than 54 years old), they were more likely to have good preventive behavior compared to the older age group: age 18–24 (AOR 2.97, 95% CI 1.68–5.25); age 25–37 (AOR 2.11, 95% CI 1.39–3.21); age 38–53 (AOR 2.04, 95% CI 1.30–3.21). Higher-income groups (>10,000 THB/month) were reported to have good preventive behavior compared to the low-income group (AOR 1.38, 95% CI 1.03–1.86). Furthermore, residents with good general knowledge of COVID-19 were 2.21 times more likely to have good preventive behavior compared to those who had poor knowledge (AOR 2.21, 95% CI 1.64–2.96).

## 4. Discussion

This study identifies the levels of knowledge, attitude, and preventive behavior toward COVID-19 in five provinces of the endemic area during the highest endemic peak in Thailand through face-to-face interviews to investigate the significant association between knowledge and attitude and the preventive behaviors toward COVID-19 in the crisis situation. The sample area of this study was located in the urban areas of central Thailand. From the findings, more than 42% of residents received a third dose of COVID-19 vaccination because COVID-19 campaigns encouraged residents in areas with high cases of COVID-19 infection to receive a robust dose of the vaccine against COVID-19 in order to create herd immunity, as promoted by the Thai government [29]. However, the number of infected residents was increasing in this area, and 53.2% of residents in this area had had COVID-19 infections because of a high level of exposure to the virus and the social–economic situation of dealing with the outbreak [19]. Suitable policies and campaigns to reduce the number of COVID-19 infections among residents in this area should be urgently implemented to reduce the number of COVID-19 infections in extreme situations. 

Thai residents in the areas of this study demonstrated low and high levels of COVID-19 knowledge in terms of general knowledge and preventive measures, respectively. Previous research [30] has identified that Thai adults show a high level of COVID-19 knowledge. These findings are generally consistent with the results of the studies conducted on Egyptian [31] and Pakistani adults [32], in which participants had a good general knowledge of the COVID-19 disease. As discussed below, knowledge of COVID-19 is closely related to COVID-19 prevention practices [26], and a high educational level could lead to high COVID-19 knowledge, resulting in an improvement in applying preventive behaviors [30]. The finding on attitude levels showed that poor levels of COVID-19 attitude among Thai residents were found in risk perception and mistrust issues. This is in contrast with another study [33] that mentioned better attitudes regarding the “new normal” guidelines among Thai people during the COVID-19 outbreak. However, the study of Kunno et al. [34] found that the level of good (50.9%) and poor (49.1%) attitudes toward COVID-19 among healthcare workers in the urban community of Bangkok, Thailand, were approximated. The reason for this result was that urban inhabitants are presumably more exposed to COVID-19, with an increasing number of COVID cases and social activities with a high risk of viral transmission in urban areas [19]. Hence, a poor attitude toward COVID-19 among Thai residents was found in endemic locations, along with a situation of panic.

However, good preventive behavior among Thai residents during the highest epidemic in Thailand is presented in this study. In terms of preventive behavior performance in COVID-19 prevention, Thai residents presented a good behavior trend, with hand washing and mask wearing in public areas, as concluded by other publications [30,31,35,36] conducted in Thailand. Various studies [37,38,39] have mentioned that broadcasts and news on television related to COVID-19 prevention can influence the preventive behaviors of residents. The Thai government enacted a policy to establish a Center for the Administrative Situation of COVID-19 and implemented many measures, such as appropriate face mask use, hand washing and clearing, social distancing, and decreasing gathering sizes, to prevent and control COVID-19 transmission within Thailand [40]. Moreover, this center promoted awareness of and preventive behavior toward COVID-19 through daily news broadcasts on television and social media to provide knowledge of the COVID-19 situation during COVID-19 transmission. The development of easy-to-understand messages throughout the social media channels of government and health sectors can be an effective strategy to reach society [41]. The results from the association between the factors and good behaviors for COVID-19 prevention by multiple logistic regression showed that different sociodemographic characteristics and other factors of residents had a significant association with preventive behaviors toward COVID-19. A discussion of each variable with significant statistics can be found below.

In analyzing sociodemographic factor variables, age and income were significantly associated with COVID-19 preventive behavior among residents during the epidemic peak in Thailand. In this study, a significant association with COVID-19 preventive behavior was found for all age groups of residents. Specifically, the younger age group (18–24 years) of residents performed preventive behavior better than other age groups and were 2.97 times more likely to be at a good level compared with poor preventive behavior toward COVID-19. This result was consistent with what Hyun et al. [42] proposed, which is that the young age group is actively involved in preventive actions with economic concerns because this age group is commonly responsible for economic activities. 

The income variable of residents was significantly associated with COVID-19 preventive behaviors in this research, consistent with various studies [33,43]. Income levels are closely related to the preventive behavior of infectious diseases, and higher-income residents can obtain better and more accurate health information than their counterparts [43]. The degree of income changes was the main variable explaining the level of difficulty experienced during the emergency. The daily life content ranked on top was also related to their economic situations (i.e., decrease in income), and those in the second and third categories were related to COVID-19 safety measures (i.e., staying home and infection prevention and control, respectively). In addition, higher-income people had more accurate knowledge of COVID-19 [43]. In contrast, McGarrity [44] found that lower-income individuals presented lower intentions to be involved in preventive behaviors [44] and a lack of knowledge of health risks [45]. The burden on household incomes could yield urban health disparities in the COVID-19 pandemic context. During the data collection in this study, the Thai government relaxed the measures of COVID-19 prevention. Residents were at their offices, wore masks, and used public transportation based on government regulations. Work-from-home conditions have been the optional way for employees. Safety products such as face masks and sanitizers are difficult for lower-income persons to obtain. In this situation, there should be a focus on socioeconomically vulnerable people when implementing the messages on or approaches to COVID-19 preventive measures. 

The current findings reveal that the residents’ general knowledge variable in this study is positively related to preventive behavior toward COVID-19. A consistent previous study in Thailand [46] mentioned that COVID-19 knowledge was found to be associated with Thai adults. In addition, these results also showed that knowledge was positively related to preventive measures, similar to a report conducted by Kamate et al. [47] and Al Ahdab et al. [48]. However, there have been no significant differences between the preventive behavior knowledge variable on the preventive behavior toward COVID-19 in multiple logistic regression analysis because the protection motivation provided useful insights into better charting people’s motivations for adapting behavioral modifications during the pandemic situation [49]. Higher knowledge was related to better individual preventive behaviors [50]. These results clearly investigate the importance of improving residents’ COVID-19 knowledge with health education, which could also result in an enhancement in their attitudes and preventive behavior towards COVID-19 [15].

The strengths of this study were, firstly, presenting the results from the large sample population recruited during the highest epidemic peak of the COVID-19 outbreak in Thailand. Compared with the other related studies in Thailand [30,33,49,51,52], the current sample size of this study was over-representative of women, well-educated residents, and working adults because stratified sampling was employed to get an equal representation of people across gender and age groups from each area [19]. Moreover, face-to-face surveys were the second strength of this study compared with other related studies in Thailand, where the samples were collected using online surveys [30,33,49,51,52,53]. Sasaki and Ichinose [19] concluded that data collection with online surveys was an important limitation of the study because residents without internet access or with a lack of internet skills were not included in the sample size of the study survey.

Several limitations of this study are presented. Firstly, a response bias may still have existed if the residents were either too stressed during the highest epidemic peak of the COVID-19 outbreak to respond or not at all stressed. This bias is similar to another study during the COVID-19 outbreak [54]. Secondly, the causal relationship of COVID-19 preventive behavior cannot be made clear by the cross-sectional survey [55]. Thirdly, COVID-19 preventive behavior has changed with the COVID-19 variants. The data of this study were collected during the highest epidemic peak of the COVID-19 outbreak. Hence, the COVID-19 preventive behavior of the residents was varied by the disease experiences of residents. Lastly, future studies with qualitative methods should be proposed to confirm the result from quantitative research, and then, the appropriate model and strategy should be recommended from the findings in both methodologies to reduce the incidence and severity of pandemic settings within a crisis situation.

## 5. Conclusions

In this research, the prediction of COVID-19 preventive behavior among Thai residents during the highest epidemic peak was identified. The finding should be recommended for the adaptation and implementation of the “new normal” and living during the pandemic. Residents were especially concerned about affecting others and others’ criticisms of them in case of infection. Using data collection and statistical analysis, we found that the young age group, high income, and high general COVID-19 knowledge were the key predictors for adopting COVID-19 preventive behavior. These findings provide intensive knowledge of COVID-19 preventive behavior in endemic areas during extreme crisis situations to support the decision process of policymakers. Rapid, formal, and reliable information on COVID-19 should be communicated by the government. Moreover, various communication and accessibility channels for the COVID-19 situation and prevention during the crisis should be given priority in order to reduce panic attacks among the Thai population at the highest peak of the pandemic.

## Figures and Tables

**Figure 1 ijerph-20-02525-f001:**
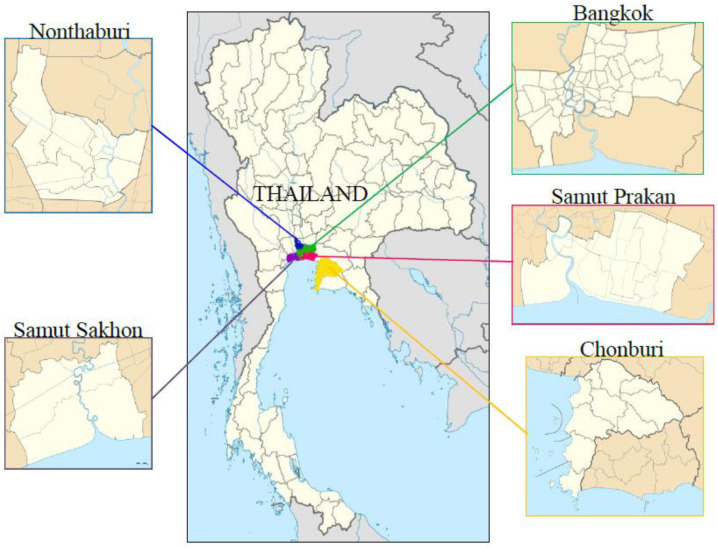
Study area.

**Figure 2 ijerph-20-02525-f002:**
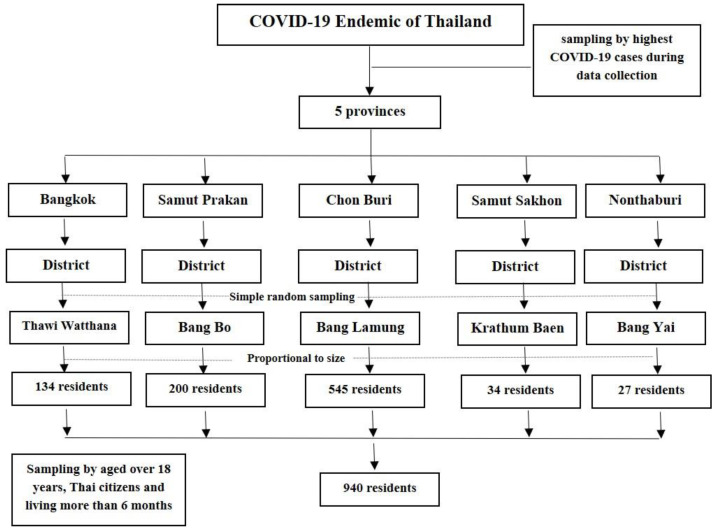
Procedures for sampling frame of resident samples.

**Table 1 ijerph-20-02525-t001:** Sociodemographic characteristics (N = 946).

Characteristics	Categories	Number, (%)
Gender	Male	376 (39.7)
	Female	570 (60.3)
Age	18–24	100 (10.6)
	25–37	286 (30.2)
	38–45	197 (20.8)
	46–53	197 (20.8)
	45+	166 (17.5)
Education	Primary school	206 (21.8)
	Secondary school	305 (32.2)
	Diploma degree	182 (19.2)
	Bachelor’s degree and above	253 (26.7)
Marital status	Single/divorced	424 (44.8)
	Married	522 (55.2)
Occupation	Self-employed	167 (17.7)
	General employee	371 (39.2)
	Student	43 (4.5)
	Government sector	88 (9.3)
	Private sector	220 (23.3)
	Farmer	8 (0.8)
	None	49 (5.2)
Income per month	Less than BHT 5000	173 (18.3)
	BHT 5001–10,000	308 (32.6)
	More than BHT 10,000	165 (49.2)
Number of vaccines received	0	91 (9.6)
	1	24 (2.5)
	2	315 (33.3)
	3	399 (42.2)
	4	177 (12.4)
Have you had a COVID-19 infection before?	Yes	503 (53.2)
	No	443 (46.8)
Source of COVID-19 infection	Do not know	99 (22.6)
	Family member	173 (39.4)
	Colleague	105 (23.9)
	High-risk area	60 (13.7)
	Other	2 (0.5)
Health insurance	Yes	178 (18.8)
	No	768 (81.2)

**Table 2 ijerph-20-02525-t002:** Knowledge and attitude levels on COVID-19 prevention.

Items and Level	Number	Percentage (%)
COVID-19 Knowledge Level		
General		
Low	121	12.8
High	825	87.2
Mean 2.72, SD 0.80, Range 1–5		
Preventive measures		
Low	329	34.8
High	617	65.2
Mean 4.69, SD 0.96, Range 1–6		
COVID-19 attitude level		
Risk perception		
Poor	507	53.6
Good	439	46.4
Mean 22.60, SD 2.90, Range 16–29		
Mistrust		
Poor	666	70.4
Good	280	29.6
Mean 21.80, SD 3.16, Range 12–30		

**Table 3 ijerph-20-02525-t003:** Knowledge of COVID-19.

Items	Yes	
	Number	%
General knowledge		
1. There is currently a symptomatic treatment cure for COVID-19.	922	97.5
2. All persons with COVID-19 will develop severe disease.	117	12.4
3. Persons with COVID-19 can transmit the virus to others.	815	86.2
4. It is not necessary for children to take measures to prevent infection by COVID-19.	130	13.7
5. The bat consumption is the risk of COVID-19 infection.	595	62.9
Preventive measures		
6. Wearing facemasks can prevent one from acquiring infection by the COVID-19 virus.	883	93.7
7. To prevent infection by COVID-19, individuals should avoid going to crowded places and avoid using public transport	683	72.2
8. Not touching face can be reduced the infection of COVID-19.	699	73.9
9. Isolating people infected with COVID-19 is an effective way to reduce the spread of the virus.	928	98.1
10. Vaccination can be reduced the severity if infected with COVID-19.	872	92.2
11. Two complete vaccinations are sufficient to prevent infection with COVID-19.	377	39.9

**Table 4 ijerph-20-02525-t004:** Attitude towards COVID-19.

Items	Strongly Disagree n (%)	Disagree n (%)	Not Sure n (%)	Agree n (%)	Strongly Agree n (%)
Risk perception					
1. I believe that I am low risk of COVID-19 infection.	77 (8.1)	179 (18.2)	465 (49.2)	149 (15.8)	76 (8.0)
2. I believe that I am low symptom if I infected COVID-19.	22 (2.3)	185 (19.6)	611 (64.6)	111 (11.7)	17 (1.8)
3. The mask wearing can be reduced the risk of COVID-19 infection.	3 (0.3)	4 (0.4)	33 (3.5)	436 (46.1)	470 (49.7)
4. The hand washing cannot be reduced the risk of COVID-19 infection.	249 (26.3)	241 (25.5)	78 (8.2)	249 (26.3)	129 (13.6)
5. The social distancing in the public area can be reduced the risk of COVID-19 infection.	1 (0.1)	8 (0.8)	36 (3.8)	474 (50.1)	427 (45.1)
Mistrust issues					
6. I am not concerned after knowing the number of COVID-19 case.	104 (11.0)	308 (32.6)	155 (16.4)	239 (25.3)	140 (14.8)
7. People who have been infected with COVID-19 should not be condemned by society.	30 (3.2)	28 (3.0)	52 (5.5)	242 (25.6)	594 (62.8)
8. When the government measures are announced, I will strictly follow.	9 (1.0)	10 (1.1)	69 (7.3)	363 (38.4)	498 (52.3)
9. Vaccination is very important.	3 (0.3)	7 (0.7)	87 (9.2)	358 (37.8)	491 (51.9)
10. The emergence of the COVID-19 is a fake	330 (34.9)	331 (32.9)	210 (22.2)	49 (5.2)	46 (4.9)
11. The COVID-19 outbreak is an attempt to reduce the world’s population.	288 (30.4)	222 (23.5)	224 (23.7)	145 (15.3)	67 (7.1)
12. People who spreads COVID-19 to others should be punished according to the law.	61 (6.4)	200 (21.1)	215 (22.7)	203 (21.5)	267 (28.2)

**Table 5 ijerph-20-02525-t005:** Preventive behavior toward COVID-19.

Items	Never n (%)	Rarely n (%)	Sometimes n (%)	Mostly n (%)	Always n (%)
1. Wearing facial masks in public areas	0 (0)	3 (0.3)	8 (0.8)	198 (20.4)	742 (78.4)
2. Keeping social distance in public areas	0 (0)	29 (3.1)	64 (6.8)	259 (27.4)	594 (62.8)
3. Washing hands frequently and using soap or hand sanitizer in the public area	2 (0.2)	14 (1.5)	44 (4.7)	288 (30.4)	598 (63.2)
4. Changing clothes before entering the house	8 (0.8)	62 (6.6)	184 (19.5)	243 (25.7)	449 (47.5)
5. Studying new information on COVID-19 prevention	16 (1.7)	112 (11.8)	206 (21.8)	278 (29.4)	334 (35.3)
6. Taking vitamin C frequently	82 (8.7)	147 (15.5)	341 (36.0)	208 (22.0)	168 (17.8)
7. Reducing travel to public areas	13 (1.4)	168 (17.8)	279 (29.5)	251 (26.5)	235 (24.8)
8. Focusing on working from home and online meetings	198 (20.9)	162 (17.1)	136 (14.4)	182 (19.2)	268 (28.3)
9. Reducing face, nose, and eye contact	14 (1.5)	30 (3.2)	150 (15.9)	343 (36.3)	409 (43.2)
10. Following all preventive behaviors above by how much	2 (0.2)	27 (2.9)	78 (8.2)	368 (38.9)	471 (49.8)

**Table 6 ijerph-20-02525-t006:** Association between factors and good preventive behaviors for COVID-19 prevention by multiple logistic regressions.

Variable	Level of Preventive Behavior		CORa (95% CI)c	*p*-Value	AORb (95% CI)c	*p*-Value
	Poor (%)	Good (%)				
**Gender**						
Male	32.7	67.3	1			
Female	36.0	70.0	1.13 (0.85–1.50)	0.378		
**Age**						
18–24	23.0	77.0	3.04 (1.74–5.30)	<0.001	2.97 (1.68–5.25)	<0.001
25–37	29.0	71.0	2.22 (1.49–3.30)	<0.001	2.11 (1.39–3.21)	<0.001
38–45	27.9	72.1	2.34 (1.51–3.62)	<0.001	2.04 (1.30–3.21)	0.002
46–53	27.4	72.6	2.40 (1.55–3.72)	<0.001	2.04 (1.30–3.21)	<0.001
54+	47.6	52.4	1		1	
**Education**						
Primary school	35.4	64.6	1			
Secondary school	35.1	64.9	1.10 (0.72–1.47)	0.934		
Diploma degree	26.9	73.1	1.49 (0.96–2.30)	0.074		
Bachelor’s degree and above	25.7	74.3	1.58 (1.06–2.34)	0.024		
**Marital status**						
Single/divorced	30.0	70.0	1.10 (0.83–1.45)	0.500		
Married	32.0	68.0	1			
Income						
<10,000	35.3	64.7	1		1	
>10,000	27.7	73.3	1.50 (1.13–1.98)	0.024	1.38 (1.03–1.86)	0.031
**Vaccines received**						
0	27.5	72.5	1			
1–2	30.4	69.6	0.86 (0.51–1.14)	0.590		
3	34.8	65.2	0.70 (0.42–1.17)	0.180		
4	23.1	76.9	1.26 (0.67–2.37)	0.468		
**COVID-19 infection**						
Yes	33.0	67.0	0.84 (0.64–1.11)	0.241		
No	29.4	70.6	1			
**Insurance**						
Yes	28.1	79.1	1.19 (0.83–1.70)	0.339		
No	31.8	68.2	1			
**COVID-19 Knowledge**						
**General**						
Low	32.1	67.9	1		1	
High	24.0	79.0	2.37 (1.71–3.02)	<0.001	2.21 (1.64–2.96)	<0.001
**Preventive measures**						
Low	42.9	57.1	1		1	
High	24.8	75.2	1.50 (0.96–2.33)	0.072	1.52 (0.96–2.40)	0.096
**COVID-19 attitude**						
**Risk perception**						
Poor	33.5	66.5	1			
Good	28.2	71.8	1.28 (0.97–1.69)	0.080		
**Mistrust**						
Poor	29.7	70.3	1			
Good	34.3	65.7	0.81 (0.60–1.09)	0.080		

## Data Availability

Not applicable.

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
