# Peer review of "Preventive Behaviors and Influencing Factors among Thai Residents in Endemic Areas during the Highest Epidemic Peak of the COVID-19 Outbreak"

_ijerph, 2023, doi:10.3390/ijerph20032525_

Round 1

Reviewer 1 Report

The COVID-19 preventive behavior and influencing factors among Thai residents during the highest epidemic peak of COVID-19. Nine hundred and forty-six residents in five districts with high COVID-19 infection cases in Thailand were systematically included in this cross-sectional survey.
I think that there are no many studies about the relation between the covid-19 preventive behaviour and the influencing factors in this region. so...it is new and original. The study of the relation between the covid-19 preventive behaviour and the influencing factors in this region. I suggested to improve the conclusions. I would talk more about the impact of the preventive measures on population and give some reccomendations for policy makers.

revise: 

line 115

line 156 it would be interesting to put the ammount in dollars.

line 162 the total of male/women is not 100% (revise)

line 215 "from" attention to the space

Author Response

Reviewer 1

I suggested to improve the conclusions. I would talk more about the impact of the preventive measures on population and give some recommendations for policy makers.

Author improve the conclusion part from our result finding and gave the recommendation to government in line 357.

line 115

Author revised the space in the word “level” in line 179.

line 156 it would be interesting to put the amount in dollars.

Author put the amount in dollars in line 171.

line 162 the total of male/women is not 100% (revise)

Author revise the total male/women for 100% in line 177.

line 215 "from" attention to the space

Author remove the space in line 232.

Reviewer 2

It seems there was a time gap between the survey data and the Highest Epidemic Peak of COVID-19 Outbreak. Could explain this time gap influenced the result or not?

After rechecking the data collection period, the data collection was during February – May 2022, which was the highest peck of COVID-19. The data from questionnaire was coded by June 2022. Hence, the period of data collection has been changed in the manuscript in line 95.

How many subjects were collected during the Highest Epidemic Peak?

Author added the information of subject number in the methodology in line 116.

How to estimate the sample size? The statstics power is ?

Author added the calculation of sample size with the power of statistic in the methodology part in line 113.

How to control the select bias?

Author added the methodology on the sample size calculation and data collection procedure for controlling the selection bias in line 120.

How to deal with the missing value?

Author added the missing data exclusion in the methodology part in line 117.

Reviewer 2 Report

The current study investigated the COVID-19 preventive behavior and their factors amonf Thai Residents. There are some comments as following:

1. It seems there was a time gap between the survey data and the Highest Epidemic Peak of COVID-19 Outbreak. Could explain this time gap influenced the result or not?

2. How many subjects were collected during the Highest Epidemic Peak?

3.  How to estimate the sample size? The statstics power is ?

4. How to control the select bias?

5. How to deal with the missing value? 

Author Response

Suggestion

Revision

Reviewer 1

I suggested to improve the conclusions. I would talk more about the impact of the preventive measures on population and give some recommendations for policy makers.

Author improve the conclusion part from our result finding and gave the recommendation to government in line 357.

line 115

Author revised the space in the word “level” in line 179.

line 156 it would be interesting to put the amount in dollars.

Author put the amount in dollars in line 171.

line 162 the total of male/women is not 100% (revise)

Author revise the total male/women for 100% in line 177.

line 215 "from" attention to the space

Author remove the space in line 232.

Reviewer 2

It seems there was a time gap between the survey data and the Highest Epidemic Peak of COVID-19 Outbreak. Could explain this time gap influenced the result or not?

After rechecking the data collection period, the data collection was during February – May 2022, which was the highest peck of COVID-19. The data from questionnaire was coded by June 2022. Hence, the period of data collection has been changed in the manuscript in line 95.

How many subjects were collected during the Highest Epidemic Peak?

Author added the information of subject number in the methodology in line 116.

How to estimate the sample size? The statstics power is ?

Author added the calculation of sample size with the power of statistic in the methodology part in line 113.

How to control the select bias?

Author added the methodology on the sample size calculation and data collection procedure for controlling the selection bias in line 120.

How to deal with the missing value?

Author added the missing data exclusion in the methodology part in line 117.
